# Egg-adaptive mutations of human influenza H3N2 virus are contingent on natural evolution

Weiwen Liang[1], Timothy J. C. Tan[2], Yiquan Wang[3], Huibin Lv[1], Yuanxin Sun[4,5], Roberto Bruzzone[1,6,7], Chris K. P. Mok[4,5]*, Nicholas C. Wu[2,3,8,9]*

**1** HKU-Pasteur Research Pole, School of Public Health, Li Ka Shing Faculty of Medicine, The University of Hong Kong, Hong Kong SAR, China, **2** Center for Biophysics and Quantitative Biology, University of Illinois at Urbana-Champaign, Urbana, Illinois, United States of America, **3** Department of Biochemistry, University of Illinois at Urbana-Champaign, Urbana, Illinois, United States of America, **4** The Jockey Club School of Public Health and Primary Care, Faculty of Medicine, The Chinese University of Hong Kong, Hong Kong SAR, China, **5** Li Ka Shing Institute of Health Sciences, Faculty of Medicine, The Chinese University of Hong Kong, Hong Kong SAR, China, **6** Istituto Pasteur Italia, Rome, Italy, **7** Centre for Immunology & Infection, Hong Kong Science Park, Hong Kong SAR, China, **8** Carl R. Woese Institute for Genomic Biology, University of Illinois at Urbana-Champaign, Urbana, Illinois, United States of America, **9** Carle Illinois College of Medicine, University of Illinois at Urbana-Champaign, Urbana, Illinois, United States of America

* kapunmok@cuhk.edu.hk (CKPM); nicwu@illinois.edu (NCW)

**Data Availability Statement:** Raw sequencing data have been submitted to the NIH Short Read Archive under accession number: BioProject PRJNA800806. Codes for analyzing next-generation sequencing data have been deposited to

## Abstract

Egg-adaptive mutations in influenza hemagglutinin (HA) often emerge during the production of egg-based seasonal influenza vaccines, which contribute to the largest share in the global influenza vaccine market. While some egg-adaptive mutations have minimal impact on the HA antigenicity (e.g. G186V), others can alter it (e.g. L194P). Here, we show that the preference of egg-adaptive mutation in human H3N2 HA is strain-dependent. In particular, Thr160 and Asn190, which are found in many recent H3N2 strains, restrict the emergence of L194P but not G186V. Our results further suggest that natural amino acid variants at other HA residues also play a role in determining the preference of egg-adaptive mutation. Consistently, recent human H3N2 strains from different clades acquire different mutations during egg passaging. Overall, these results demonstrate that natural mutations in human H3N2 HA can influence the preference of egg-adaptation mutation, which has important implications in seed strain selection for egg-based influenza vaccine.

## Author summary

Growing influenza virus in chicken eggs remains a widely used method for producing seasonal influenza vaccines. However, it is common for influenza virus to acquire mutations on the hemagglutinin (HA) protein to facilitate their growth in eggs. These egg-adaptive mutations may change the antigenic property of HA and thus decrease the vaccine effectiveness, as exemplified by mutation L194P. In this study, we demonstrate that two natural variants Thr160 and Asn190 in recently circulating human H3N2 strains can prevent the emergence of L194P. Consistently, we also show that the preference of egg-adaptive

https://github.com/Wangyiquan95/HA_egg_passage. Code and source files for structural modeling using Rosetta have been deposited to https://github.com/timothyjtan/HA-RBD_height.

**Funding:** This work was supported by a fellowship from the Pasteur Foundation Asia (W.L.), the Calmette and Yersin scholarship from the Pasteur International Network Association (H.L.), the funding from Centre for Immunology & Infection, InnoHK, an initiative of the Innovation and Technology Commission, the Government of the Hong Kong SAR (R.B.), the startup fund from the University of Illinois at Urbana-Champaign (N.C. W.), the Health and Medical Research Fund (no.19180932) (C.K.P.M.), the National Research Foundation of Korea (NRF) grant funded through the Korea government (NRF-2018M3A9H4055203) (C.K.P.M.), Guangdong-Hong Kong-Macau Joint Laboratory of Respiratory Infectious Disease (20191205) (C.K.P.M.) and the visiting scientist scheme from Lee Kong Chian School of Medicine, Nanyang Technological University, Singapore (C.K. P.M.). The funders had no role in study design, data collection and analysis, decision to publish, or preparation of the manuscript.

**Competing interests:** The authors have declared that no competing interests exist.

mutation varies from strains to strains, with various previously uncharacterized egg-adaptive mutations in recent human H3N2 clades. These observations indicate that the preference of egg-adaptive mutation changes as human influenza virus evolves over time. Overall, our findings provide valuable insights into the mechanism of egg-adaptation of influenza virus and have important implications for the development of more effective influenza vaccines.

## Introduction

Vaccination is considered the most effective approach to prevent influenza infection. Seasonal influenza vaccine offers protection against two influenza A subtypes, namely H1N1 and H3N2, as well as influenza B viruses. However, the vaccine effectiveness against H3N2 is typically lower than H1N1 and influenza B [1–3]. Besides vaccine mismatch in certain influenza seasons, the occurrence of egg-adaptive mutations during vaccine production can also lead to the reduction of vaccine effectiveness [4–6]. When grown in eggs, human H3N2 viruses often acquire mutations in the receptor-binding site (RBS) of hemagglutinin (HA) to facilitate the binding to α2,3-linked sialic acid, which is the prototypical receptor for avian influenza viruses [7–9]. Since the HA RBS partially overlaps with several major antigenic sites [10], some egg-adaptive mutations can alter the antigenicity of HA [4, 5, 9, 11–13].

G186V and L194P are the two most common egg-adaptive mutations in H3N2 HA [14]. Although both mutations are in the RBS, L194P but not G186V significantly alters the antigenicity of HA [15]. Thus, identifying factors that influence the preference of egg-adaptive mutation can help optimize the effectiveness of egg-based influenza vaccine. Our previous study has demonstrated a strong incompatibility between G186V and L194P, in which their co-occurrence is extremely deleterious to the virus [14]. This epistatic interaction suggests that G186V and L194P represent two mutually exclusive egg-adaptive mutations. Given that epistasis is pervasive in the HA RBS and that HA RBS is constantly evolving [13,16–19], it is possible that different strains have different preferences of egg-adaptive mutation. However, it remains to be explored whether such differential preference exists.

In this study, sequence analysis and mutagenesis experiments are used to identify amino acid variants on human H3N2 HA that can affect the preference of egg-adaptive mutation. We discovered that L194P is incompatible with T160 and N190, both of which are prevalent in recent human H3N2 strains. Consistently, recent human H3N2 strains are incompatible with L194P. Our results further indicate that amino acid variants at other residues also contribute to the L194P incompatibility in recent human H3N2 strains. Along with the observation that human H3N2 strains from different clades prefer different egg-adaptive mutations, we conclude that the preference of egg-adaptive mutation changes as human H3N2 virus evolves.

## Results

### Differential egg-adaptation preferences in historical vaccine strains

To examine whether different human H3N2 strains have different preferences of egg-adaptive mutations, HA sequences from egg-passaged strains were collected from the Global Initiative for Sharing Avian Influenza Data (GISAID) [20] (**S1 Table**). Specifically, we focused on egg-passaged strains that were derived from WHO-recommended candidate vaccine viruses for influenza seasons between 2008 and 2023. Egg-passaged strains that were derived from different parental viruses have acquired different egg-adaptive mutations (**Fig 1A**). For example, all

egg-passaged strains from A/Victoria/361/2011, A/Texas/50/2012, A/Switzerland/9715293/2013, and A/Hong Kong/2671/2019 have acquired G186V, whereas those from A/Hong Kong/4801/2014, A/Singapore/INFIMH-16-0019/2016 (Sing16) and A/Switzerland/8060/2017 (Switz17) have acquired L194P. This observation is consistent with the hypothesis that the preference of egg-adaptive mutation is strain dependent.

To further test our hypothesis, we measured the replication fitness of G186V and L194P in Sing16, Switz17, and A/Kansas/14/2017 (Kansas17) (**S2 Table**). Our mutagenesis experiments were based on egg-adapted strains. As a result, all the Kansas17 mutants in this experiment were constructed in the presence of D190N and S219Y, which were present in 100% (15/15) and 87% (13/15), respectively, of egg-passaged Kansas17 strains in GISAID. Similarly, all the Sing16 and Switz17 mutants were constructed in the presence of T160K, which was found in all egg-passaged strains derived from Sing16 and Switz17 in GISAID and could alter the antigenicity of the virus (**Fig 1B**) [5]. The fitness effects of four different combinations of amino acid variants at residues 186 and 194 were tested, namely 1) G186/L194, which represented the amino acid variants in the unpassaged strains, 2) V186/L194, which represented the single egg-adaptive mutation G186V, 3) G186/P194, which represented the single egg-adaptive mutation L194P, and 4) V186/P194, which represented the co-occurrence of two incompatible egg-adaptive mutations [14].

As expected, all variants with G186/L194 could be rescued, whereas all variants with V186/P194 had no detectable titer (**Fig 1C**). Furthermore, all rescuable variants showed comparable replication kinetics in hMDCK cells, despite G186/P194 had a slightly lower titer in Switz17 and V186/L194 had a slower growth in Kansas17 (**S1A Fig**). In contrast, the fitness effects of V186/L194 and G186/P194 in eggs differed dramatically among strains. For example, V186/L194 only facilitated the egg-adaptation of Kansas17-N190/Y219, but not Sing16-K160 and Switz17-K160 (**Fig 1D**). Similarly, G186/P194 could only be rescued in Sing16-K160 and Switz17-K160, but not Kansas17-N190/Y219 (**Fig 1C**). Next-generation sequencing further showed that during egg-passaging, V186/L194 were stable in Kansas17-N190/Y219, and G186/P194 were stable in both Sing16-K160 and Switz17-K160 (**Fig 1E and S3 Table**. These results substantiate that there are differential preferences of egg-adaptive mutations among different strains.

## Residues 160 and 190 influence the egg-adaptation preference

The inability of rescuing G186/P194 in Kansas17-N190/Y219 suggests that G186V is not the only mutation that is incompatible with L194P. We therefore aimed to identify additional mutations that were incompatible with L194P. The HA sequence of Kansas17 X-327, which was an egg-adapted strain equivalent to the Kansas17-V186/N190/L194/Y219 (see above, **Fig 1C**), was compared to that of Switz17 NIB-112 and Sing16 NIB-104, which were two egg-adapted strains with K160/P194. Of note, the HA sequences are equivalent between Kansas17 X-327 with V186G/L194P (**Fig 2A**) and the unrescuable Kansas17-G186/N190/P194/Y219 (see above, **Fig 1C**). This sequence analysis led us to identify four candidate amino acid variants in the RBS that might be incompatible with L194P, namely S159, N190, S193 and Y219. Subsequently, S159Y, N190D, S193F and Y219S, which represented the RBS mutations from Kansas17 X-327 to Switz17 NIB-112 and Sing16 NIB-104, were introduced individually into the HA of Kansas17 X-327 with V186G/L194P. V186G was included in since V186 and P194 were known to be incompatible [14]. Our virus rescue experiment showed that the fitness defect of V186G/L194P in Kansas17 X-327 could be restored by N190D (**Fig 2B**). Interestingly, although Kansas17 X-327 with V186G/L194P/N190D had a lower replication fitness in hMDCK cells compared to X-327, it showed comparable replication fitness with X-327 in eggs (**Figs 2C and S1B**). Next-generation sequencing further confirmed that Kansas17 X-327 with

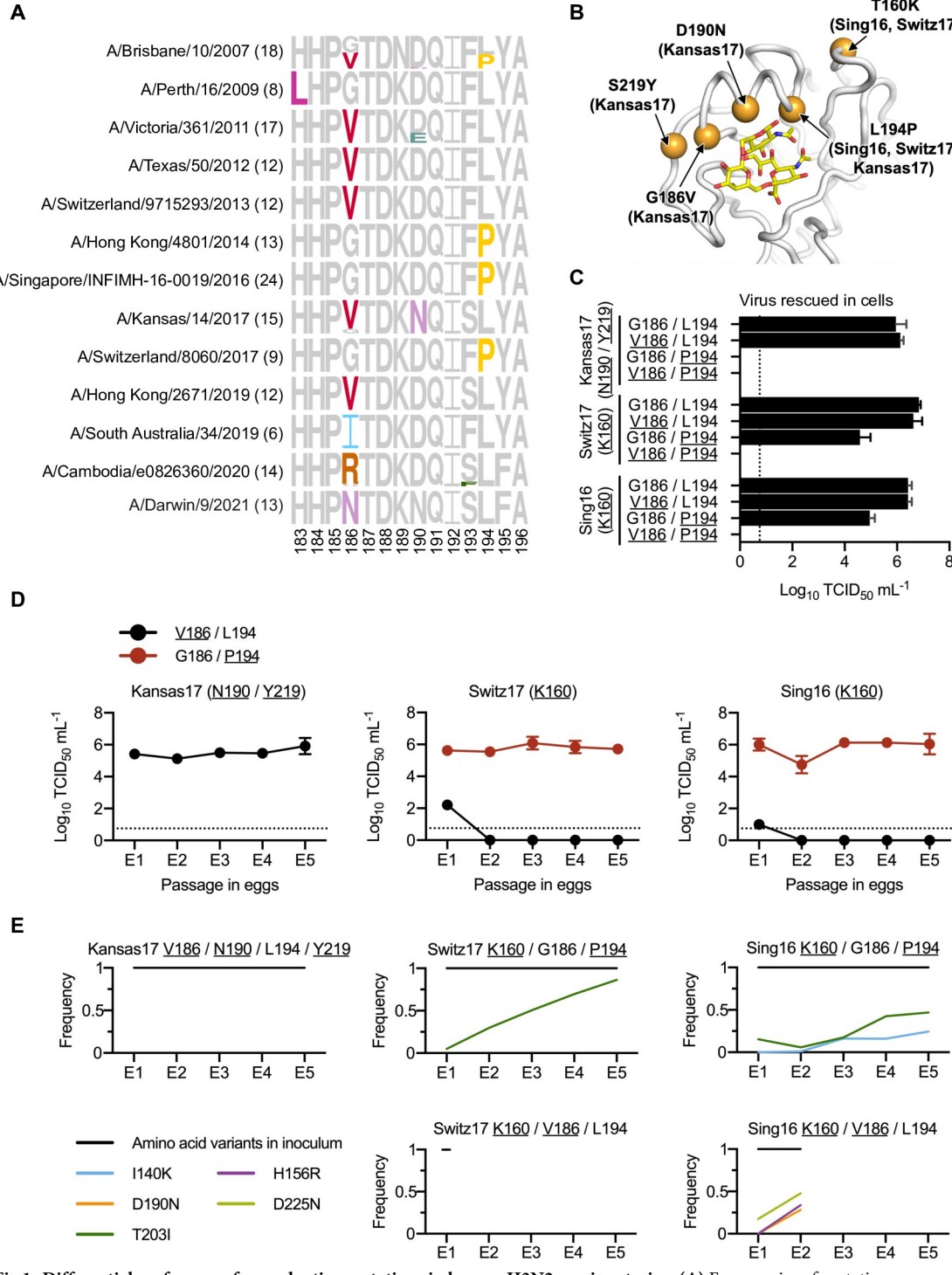

**Fig 1. Differential preference of egg-adaptive mutations in human H3N2 vaccine strains. (A)** Frequencies of mutations among different egg-passaged seasonal H3N2 vaccine strains are shown as sequence logos. Amino acid variants from residues 183 to 196 are shown (H3 numbering). The number of egg-passaged strains included in this analysis is indicated in the parenthesis. The relative size of each amino acid letter represents its frequency in the sequences. Grey letters represent the amino acid variants that are observed in the corresponding unpassaged parental strains. **(B)** Key egg-adaptive mutations in this study are shown in orange (PDB 6BKT) [17]. Sialylated glycan receptor is in yellow sticks representation. Strains in parentheses are associated with the corresponding egg-adaptive mutations. Only three strains of interest (Sing16, Switz17, and Kansas17) are included here. **(C)** Replication fitness of different mutants of human H3N2 vaccine strains was examined in a virus rescue experiment in mammalian cells (293T/hMDCK). Viral titers were measured by TCID$_{50}$. **(D)** The viral titer of each human H3N2 vaccine strain with either

V186/L194 or G186/P194 was measured during egg-passaging. **(C-D)** The means of three independent experiments are shown with SD indicated by the error bars. The dashed line represents the lower detection limit. Amino acid variant representing an egg-adaptive mutation is underlined. **(E)** Frequencies of mutations in the receptor-binding subdomain (residues 117–265) [21] that emerged during serial passaging in eggs. The strain of inoculum for each passaging experiment is indicated above each plot, with those representing egg-adaptive mutations underlined. Frequencies are shown as means of three biological replicates. Only those mutations that reached a minimum average frequency of 10% after the fifth passage are plotted.

V186G/L194P/N190D was genetically stable during egg-passaging (**Fig 2D and S3 Table**). To examine if the incompatibility between N190 and P194 could be generalized to other strains, we further introduced D190N into Switz17-K160/P194 and Sing16-K160/P194, which were equivalent to Switz17 NIB-112 and Sing16 NIB-104, respectively. Consistent with the results for Kansas17 X-327, D190N was deleterious in Switz17-K160/P194 and Sing16-K160/P194 (**Fig 2E**). Together, these results demonstrate the incompatibility between N190 and P194.

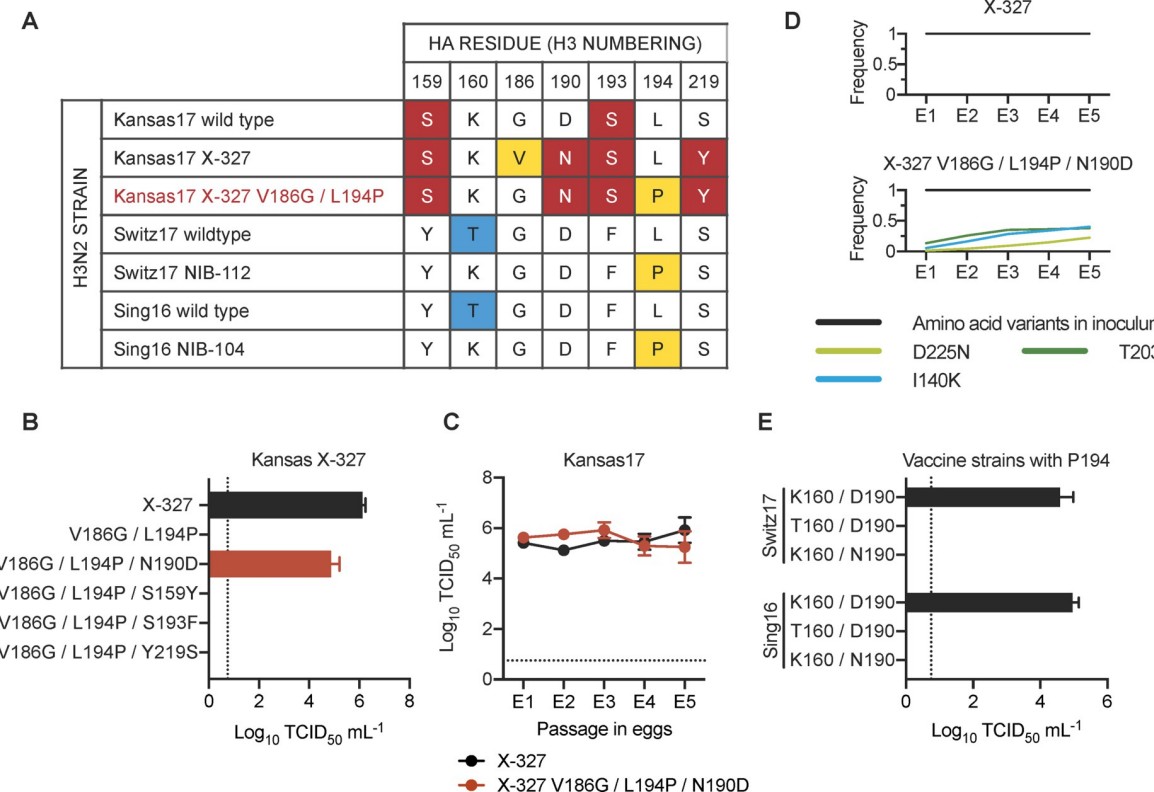

**Fig 2. Epistatic interactions that involve egg-adaptive mutations. (A)** Amino acid variants from residues 150 to 220 among the indicated strains are shown. The rest of the sequences from residues 150 to 220 were completely conserved among these strains. Major egg-adaptive mutations G186V and L194P are in yellow. Amino acid variants that are unique to Kansas17 strains are in red while variants unique to Switz17 and Sing16 are in blue. Of note, Kansas17 X-327, Switz17 NIB-112, and Sing16 NIB-104 are egg-adapted strains, whereas Kansas17 X-327 V186G/L194P (red) was a mutant generated in this study. **(B)** Replication fitness of Kansas17 X-327 with indicated mutations was examined in a virus rescue experiment in mammalian cells (293T/hMDCK). **(C)** The replication fitness of Kansas17 X-327 and Kansas17 X-327 with triple mutations (V186G/L194P/N190D) in eggs were examined during serial egg-passaging. **(D)** Frequencies of mutations in the receptor-binding subdomain (residues 117–265) [21] that emerged during egg-passaging of the indicated strains. Results are shown as means of three independent biological replicates. Only those mutations that reached a minimum average frequency of 10% after the fifth passage are plotted. **(E)** Replication fitness of Sing16 and Switz17 L194P virus with or without K160T and D190N mutations was assessed in a virus rescue experiment in mammalian cells (293T/hMDCK). **(B, C, E)** All viral titers were measured by $TCID_{50}$. The means of three independent experiments are shown with SD indicated by the error bars. The dashed line represents the lower detection limit.

During our sequence analysis, we noticed that the egg-adaptive mutation T160K [5] often co-occurred with L194P, as exemplified by Switz17 NIB-112 and Sing16 NIB-104. Our rescue experiment showed that T160 was incompatible with P194 in both Sing16 and Switz17 (**Fig 2E**). As a result, in addition to V186 and N190, T160 is another amino acid variant that is incompatible with P194.

## Recent human H3N2 strains are incompatible with L194P

Among the amino acid variants that are incompatible with P194, T160 and N190 have reached high occurrence frequency in recently circulating human H3N2 strains. T160, which introduced an N-glycosylation site at N158, was fixed in clade 3C.2a during the 2014–2015 influenza season [5], whereas N190 emerged more recently along with I160 and D186 and reached >90% in 2021 (**Fig 3A**). Not surprisingly, A/Victoria/22/2020 (Vic20) and A/Italy/11871/2020 (Italy20), which were both from clade 3C.2A1b.1a and had T160/N190, could not be rescued in the presence of L194P (**Fig 3B**). In contrast, introducing V186 into Italy20 yielded high titers in both virus rescue and egg-passaging experiments (**Fig 3C**). These observations illustrate that recent human H3N2 strains have a strong preference against L194P as determined in the egg-adapted viruses.

While we have shown above that T160 and N190 were incompatible with L194P in Kansas17, Switz17, and Sing16 (**Fig 2**), Italy20 with T160K/N190D was still incompatible with L194P (**Fig 3D**). This result indicates that besides T160 and N190, additional amino acid variants in Italy20 were also incompatible with L194P. We postulated that those additional amino acid variants could be identified by comparing the HA sequences of Italy20 and Sing16, since Sing16 was compatible with L194P in the presence of K160/D190 (**Fig 2E**). Eight mutations in the receptor binding subdomain (residues 117–265) [21] were identified between Sing16 and Italy20 (**Fig 3E**). Based on spatial proximity between these eight mutations and L194P, we hypothesized that the differential fitness effects of L194P in Sing16 and Italy20 could be due to the presence of mutations G186D, F193S, and S198P in Italy20. Since the compatibility with L194P in Kansas17 was not influenced by the presence of S193 as shown in our rescue experiment above (**Fig 2A and 2B**), we focused on D186 and P198 here. However, even with mutations D186G and P198S, Italy20-T160K/D186G/N190D/P198S/L194P still could not be rescued (**Fig 3D**). As demonstrated in our previous study, G186V significantly increases the RBS height [14]. In contrast, L194P decreases the RBS height. The co-existence of these two mutations with opposite structural effects destabilizes the 190-helix and abolishes receptor binding [14]. However, structural modeling showed that none of the eight mutations increased the height of the RBS (**Fig 3F**). Together, these results demonstrate that the L194P incompatibility in recent human H3N2 strains is conferred by multiple amino acid variants in addition to T160 and N190, and likely involves a mechanism that is different from G186V.

## Egg-adaptation preference is clade dependent

The recent emergence of L194P incompatibility suggests that the preference of egg-adaptive mutation has been evolving. To understand the preference of egg-adaptation in recent human H3N2 strains, we analyzed the HA sequences of 72 egg-passaged H3N2 strains from years 2019 to 2021, which belong to antigenically distinct clades [22–24]. By comparing the HA sequences between the egg-passaged strains and their non-egg-passaged counterparts, we observed that egg-adaptive mutations often emerged at residues 160, 186, 190, 194, 219 and 225 (**Fig 4** and **S4 Table**). In addition, different clades showed different preferences of egg-adaptive mutation. This differential preference could be exemplified by residues 186 and 225. While all non-egg-passaged strains from both clades 3C.2a1b.1a and 3C.2a1b.2a2 had D186,

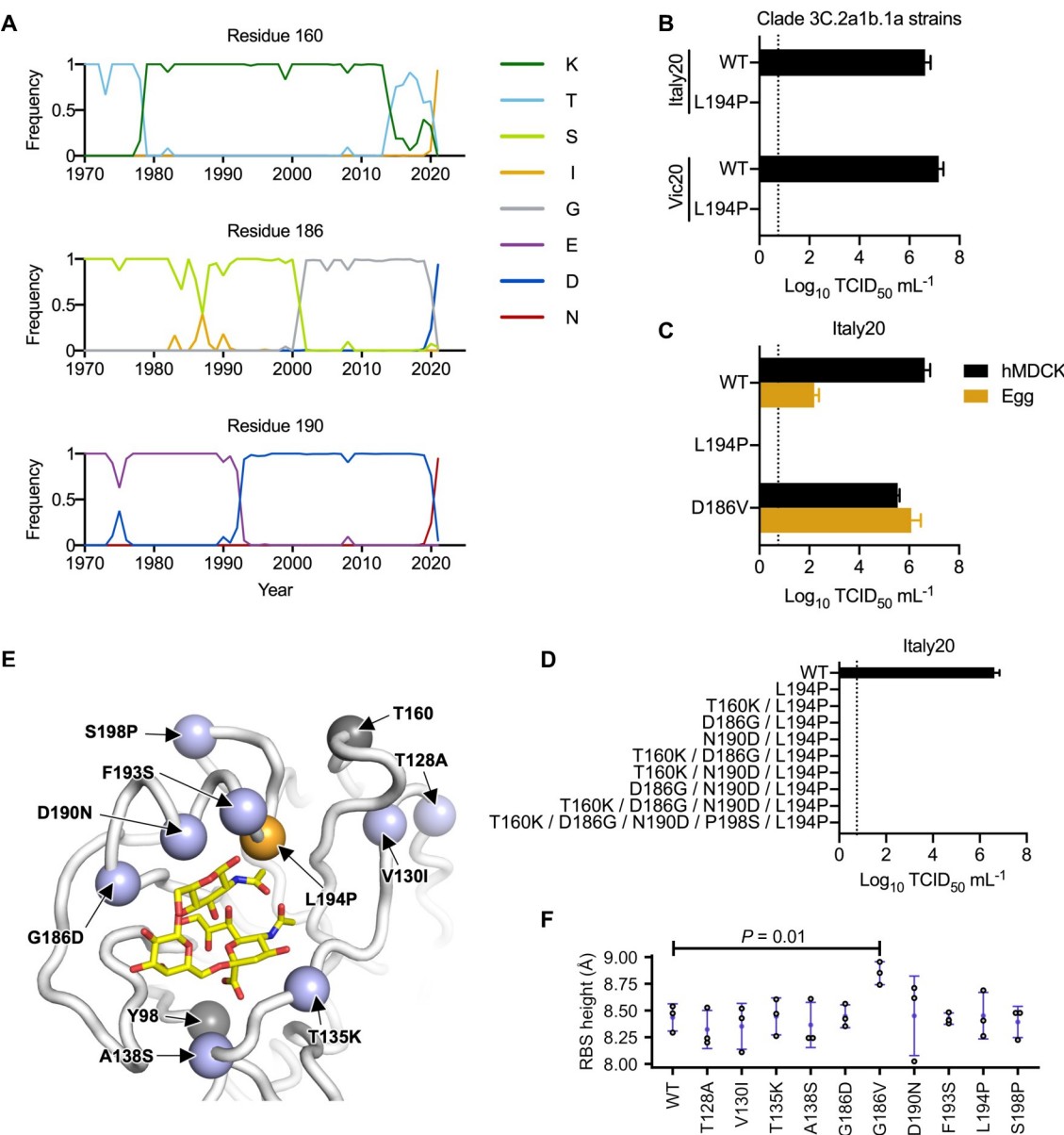

**Fig 3. Incompatibility between N190 and P194. (A)** Frequencies of amino acid variants at residues 160, 186 and 190 in human H3N2 HA over time are shown. Only those variants that reached a maximum annual frequency of 35% are plotted. **(B)** Two wild type (WT) strains in clade 3C.2a1b.1a (Vic20 and Italy20) with or without L194P mutation were examined in a virus rescue experiment in mammalian cells (293T/hMDCK). Of note, most unpassaged strains in clade 3C.2a1b.1a, including both Vic20 and Italy20, naturally contain N190. **(C)** The replication fitness of Italy20 with indicated mutations was examined in a virus rescue experiment in mammalian cells (293T/hMDCK) as well as in an egg-passaging experiment, in which $10^4$ $TCID_{50}$ of the rescued viruses were propagated in eggs for 48 hours. **(D)** The replication fitness of Italy20 with indicated mutations was measured in a virus rescue experiment in mammalian cells (293T/hMDCK). **(B-D)** Viral titers were measured by $TCID_{50}$. The means of three independent experiments are shown with SD indicated by the error bars. The dashed line represents the lower detection limit. **(E)** The locations of mutations between Sing16 and Italy20 in the receptor-binding subdomain (residues 117–265) [21] are shown in blue (PDB 6BKT). The locations of T160 and Y98 are shown in grey [17]. L194P is in orange. Sialylated glycan receptor is in yellow sticks representation. **(F)** The structural impact of each mutation was modeled by Rosetta [48]. The distance between the phenolic oxygen of Tyr98 ($OH_{98}$) and the $C\alpha$ of residue 190 was computed as the height of receptor-binding site (RBS). Three replicates, each with 100 simulations, were performed. Each data point represents the lowest scoring pose in a replicate. Error bars represent the SD. *P*-value was computed by two-tailed t-test. Only the difference between WT and G186V is statistically significant (*P* < 0.05).

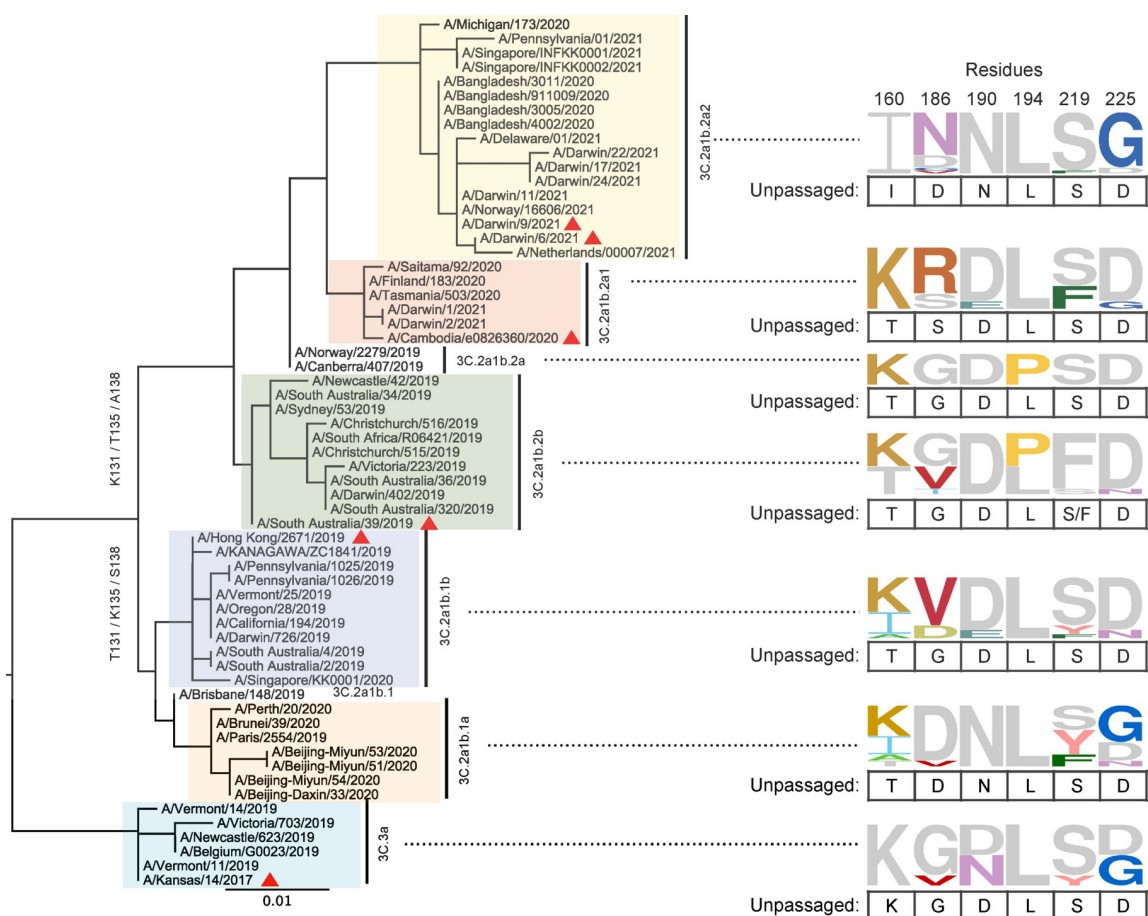

**Fig 4. Egg-adaptive mutations in recent human H3N2 HA.** A rooted phylogenetic tree was built using maximum likelihood (ML) method with 100 bootstraps on the HA sequences of 61 human H3N2 strains without egg-passaging (unpassaged strains). Different clades are highlighted in different colors. WHO-recommended vaccine strains are indicated by a red triangle symbol. These unpassaged human H3N2 strains correspond to the parental strains of 72 egg-passaged strains. Since multiple egg-passaged strains with different HA sequences could be generated from a single unpassaged strain, the number of unpassaged strains is less than that of egg-passaged strains. Frequencies of amino acids at HA residues 160, 186, 190, 194, 219 and 225 among egg-passaged strains in different clades are shown as sequence logos. The relative size of each amino acid letter represents its frequency in the sequences. For each clade, amino acid variants that are also observed in the unpassaged strains are in grey and listed below each sequence logo.

egg-adaptive mutation D186N was common in 3C.2a1b.2a2 but not observed in 3C.2a1b.1a. Similarly, although D225 was present in all non-egg-passaged strains across all clades, egg-adaptive mutation D225G was prevalent in 3C.2a1b.2a2 but not observed in clades 3C.2a1b.2a, 3C.2a1b.2b, and 3C.2a1b.1b. These observations are consistent with the notion that the preference of egg-adaptive mutation in human H3N2 HA is influenced by natural mutations.

## Discussion

Egg-adaptation of influenza virus has been central to the production of seasonal influenza vaccine. Our work here shows that the preference of egg-adaptive mutation is strain dependent, due to epistasis between egg-adaptive mutations and natural amino acid variants. Specifically, we identified two natural amino acid variants T160 and N190 that are incompatible with the egg-adaptive mutation L194P. Our results also indicate that additional amino acid variants are responsible for the L194P incompatibility in recent human H3N2 HA. Besides L194P, many other egg-adaptive mutations, albeit at a lower prevalence, were observed in historical vaccine

strains, such as H156Q, H156R, H183L and A196T [14,25–27]. The sequence determinants that influence the preference of other egg-adaptive mutations are currently unclear. Comprehending the epistatic interactions that involve egg-adaptive mutations as well as the underlying mechanisms will be key to accurately predict mutations that arise during egg-adaptation.

Our previous studies have shown that epistasis is pervasive between mutations in the HA RBS [16–18]. One interesting observation is that epistasis between two HA RBS mutations can be modulated by a third mutation. For example, epistasis between the engineered mutations L226R and S228E in an H1 HA could only be observed in the presence of G225E [16]. This type of higher-order epistasis may play a role in determining the preference of egg-adaptive mutation. Despite the incompatibility between T160 and P194 in Sing16 and Switz17 (**Fig 2E**), we noticed the co-occurrence of T160 and P194 in four egg-adapted H3N2 strains (EPI1588456, EPI1584616, EPI1526498 and EPI1440496), which all belong to clade 3C.2a1b.2b (**Fig 4 and S4 Table**). This observation implies that the incompatibility between T160 and P194 can be masked by other amino acid variants. As shown by a recent study, epistasis can also alter the antigenic effect of a given mutation in the HA RBS [28]. Whether the antigenic effects of egg mutations are influenced by epistasis will need to be explored in future studies.

We anticipate that the preference of egg-adaptive mutation will keep changing as human H3N2 HA continues to evolve. As a result, future studies of egg-adaptive mutations in human H3N2 strains are warranted. Currently, 3C.2a1b.2a2 is the dominant clade of circulating human H3N2 virus. Although our data show that egg-adaptive mutation L194P is unlikely to occur in clade 3C.2a1b.2a2, egg-adaptive mutations D186N and D225G are common in this clade [15]. However, the antigenic effects of D186N and D225G in clade 3C.2a1b.2a2 remain to be explored. Given that different egg-adaptive mutations can have different antigenic effects [5,8,9,12,15,24–32], systematic characterization of individual egg-adaptive mutations is critical for the selection of vaccine seed strains with minimal antigenic change. Nevertheless, egg-based influenza vaccine production is an 80-year-old technology that needs to be replaced by more advanced alternatives. An important step forward was the commercialization of cell-based and recombinant influenza vaccines, which have shown better effectiveness than egg-based influenza vaccines [33,34]. In addition, a Phase 1/2 clinical trial of an mRNA seasonal influenza vaccine candidate has initiated recently [35]. Despite the potentially slow transition, these alternatives will provide an ultimate solution to the problems of egg adaptation.

## Materials and methods

### Sequence analysis

Full-length HA sequences of WHO-recommended H3N2 candidate vaccine virus strains from 2008 to 2022 were downloaded from the Global Initiative for Sharing Avian Influenza Data (GISAID; http://gisaid.org) [20]. Information for these sequences is shown in **S1 Table**. Information for the full-length HA sequences of viruses from 2019 to 2021, with and without egg-passaging, is shown in **S4 Table**. Sequences were aligned by MAFFT version 7 (https://mafft.cbrc.jp/alignment/software/) with default setting [36]. Passaging history was determined by parsing the regular expression in FASTA headers as described [37]. The frequencies of amino acids were plotted using WEBLOGO (https://weblogo.berkeley.edu/logo.cgi) [38]. Phylogenetic tree was built and rooted using maximum likelihood (ML) method by Geneious Prime (Geneious), with 100 bootstraps.

### Cell culture

Humanized Madin-Darby canine kidney (hMDCK) cells with higher and stable expression of human 2,6-sialtransferase were kindly provided by Professor Yoshihiro Kawaoka, the

University of Tokyo [39]. Both hMDCK cells and human embryonic kidney (HEK) 293T cells were maintained in minimal essential medium (MEM) supplemented with 10% fetal bovine serum, 25 mM HEPES, and 100 U mL$^{-1}$ penicillin-streptomycin (PS).

## Virus rescue experiments

All H3N2 viruses generated in this study were based on the influenza eight-plasmid reverse genetics system [40,41]. The HA and neuraminidase (NA) genes of the strains of interest (S2 Table) were synthesized by Sangon Biotech, and cloned into the pHW2000 vector as previously described [14]. For HA, the ectodomain was from the strains of interest, whereas the non-coding region, N-terminal secretion signal, C-terminal transmembrane domain, and cytoplasmic tail were from H1N1 A/PR/8/34 (PR8). For NA, the entire coding region was from the strains of interest, whereas the non-coding region of NA was from PR8. Mutations were introduced into the constructs by polymerase chain reaction (PCR). Primers used in this study were produced by Integrated DNA Technologies. Recombinant chimeric 6:2 reassortant viruses with the six internal genes (PB2, PB1, PA, NP, M, and NS) from PR8 were rescued by transfecting a co-culture of HEK 293T and hMDCK cells at a 6:1 ratio with 70% confluence in 6-well plate. Transfection was performed using 16 μL of TransIT-LT1 (Mirus Bio) and 1 μg each of the eight plasmids encoding the corresponding influenza virus segments (a total amount of 8 μg of DNA). At 6 h post-transfection, medium was replaced with 1 mL of MEM. At 24 h post-transfection, another 1 mL of MEM supplemented with 1 μg mL$^{-1}$ tosylphenylalanyl chloromethyI ketone (TPCK)-trypsin was added. Supernatant was harvested at 72 h post-transfection, all of which was further inoculated into hMDCK cells at 90% confluence in a T75 flask. Supernatant of the hMDCK cells was harvested when more than 70% of cells had cytopathic effect (CPE). If less than 70% of cells had CPE by 72 h post-infection, cell supernatant was harvested regardless. Virus titer was determined by titration in hMDCK cells as previously described [42,43]. For each mutant, three independent rescue experiments were performed. All our rescue experiments included a positive control that contained wild type (WT) HA and a negative control that contained no HA. Viral RNAs were extracted, reverse transcribed by Superscript III reverse transcriptase (Thermo Fisher Scientific), and sequenced by Sanger sequencing to confirm its sequence integrity before passaging in eggs.

## Multicycle viral growth kinetics

To measure the viral multicycle viral growth kinetics, rescuable viruses were inoculated into hMDCK cells at an MOI of 0.01, and supernatants were harvested at 24, 48 and 72 h post-infection [44]. Viral titers in the supernatants were determined by titration in hMDCK cells as previously described [42,43].

## Virus passaging in embryonic eggs

Embryonic eggs at 10 days post-fertilization were used for the virus passaging. For each passage, $10^4$ TCID$_{50}$ of viruses were inoculated into one egg. The only exception was for the next-generation sequencing experiment (see below), $10^6$ TCID$_{50}$ was used for the first passage of Sing16-K160/V186/L194. The allantoic fluid was harvested at 48 h post-infection. Virus titer was determined by titration in hMDCK cells as previously described [42,43]. Each experiment consisted of five consecutive passaging and was performed in triplicate.

## Next-generation sequencing of egg-passaged viruses

The viral RNA of the egg-passaged virus was extracted using QIAamp Viral RNA Mini Kit (QIAGEN). The extracted RNA was then reverse transcribed to cDNA using ProtoScript II

First Strand cDNA Synthesis Kit (New England Biolabs). Subsequently, PCR was performed using PrimeSTAR Max DNA Polymerase (Takara Bio) with the cDNA as template and virus-specific primers 5'-CAC TCT TTC CCT ACA CGA CGC TCT TCC GAT CT NNN GGT CAC TAG TTG CCT CAT CCG G-3' and 5'- GAC TGG AGT TCA GAC GTG TGC TCT TCC GAT CTG GTG CAT CTG ATC TCA TTA TTG-3'. A second round of PCR was performed to add the rest of adapter sequence and index to the amplicon using primers 5'- AAT GAT ACG GCG ACC ACC GAG ATC TAC ACT CTT TCC CTA CAC GAC GCT-3' and 5'- CAA GCA GAA GAC GGC ATA CGA GAT XXX XXX GTG ACT GGA GTT CAG ACG TGT GCT-3'. Nucleotides in primers annotated by an 'N' or "X" represented the index sequence for distinguishing PCR products derived from different viruses, passages, and biological replicates. The final PCR products were sequenced by Illumina MiSeq PE300.

## Analysis of next-generation sequencing data

Sequencing data was obtained in FASTQ format and analyzed using in-house Python and shell scripts. Briefly, paired-end reads were merged by PEAR [45]. The merged reads were then parsed by SeqIO module in BioPython and translated into protein sequences [46]. For each sample, the frequency of amino acid variant $i$ was computed as follows:

$$frequency_i = \frac{readcount_i}{sequencingdepth}$$

where $readcount_i$ represents the number of reads that contain amino acid variant $i$ and $sequencingdepth$ represents the total number of reads in the sample of interest.

## Structural modeling

Residues 57 to 261 of a monomer of A/Michigan/15/2014 (H3N2) influenza virus HA (PDB 6BKT) [17] was extracted. The PDB file was renumbered using pdb-tools [47]. Point mutagenesis for T128A, V130I, T135K, A138S, G186D, G186V, D190N, F193S, L194P and S198P (all in original numbering) were performed using the fixed backbone (fixbb) design application in Rosetta (RosettaCommons). For each replicate, one hundred poses were generated and the lowest scoring pose was used for downstream analysis. The fixbb application for each mutation was performed in triplicate. A constraint file for each lowest scoring pose was obtained using the minimize_with_cst application in Rosetta and converting the log file to a constraint (cst) file using the convert_to_cst_file application in Rosetta. Subsequently, fast relax was performed on wild type or mutant using the relax application in Rosetta with the corresponding constraint file [48]. For each replicate, 30 poses were generated and the lowest scoring pose was used to measure the height of the RBS, which is the distance between the side chain oxygen atom of Y98 (H3 numbering) and Cα of amino acid 190 (H3 numbering) using PyMOL (Schrödinger).

## Supporting information

**S1 Fig. Viral replication kinetics of different H3N2 variants in hMDCK cells.** (A) Viral replication kinetics of rescuable mutants of H3N2 vaccine strains were examined in hMDCK cells. (B) Viral replication kinetics of Kansas17 X-327 and Kansas17 X-327 with V186G/L194P/ N190D were examined in hMDCK cells. (A, B) hMDCK cells were inoculated with the indicated variant at an MOI of 0.01. Viral titers in supernatants harvested at 24, 48 and 72 h post-infection were measured by TCID50 using hMDCK cells. The means of three independent experiments are shown with SD indicated by the error bars. The dashed line represents the lower detection limit. Amino acid variant representing an egg-adaptive mutation is underlined. (TIF)

**S1 Table. H3N2 components of WHO-recommended influenza vaccines for influenza seasons between 2008 and 2023.**
(DOCX)

**S2 Table. H3N2 genes that were used in this study.**
(DOCX)

**S3 Table. Frequencies of mutations in egg-passaged viruses that showed an average frequency of >10% in the fifth passage.**
(DOCX)

**S4 Table. Information of HA from human H3N2 strains with or without egg-passaging.**
(DOCX)

## Author Contributions

**Data curation:** Yiquan Wang.

**Formal analysis:** Weiwen Liang, Timothy J. C. Tan, Yiquan Wang, Huibin Lv.

**Funding acquisition:** Weiwen Liang, Huibin Lv, Chris K. P. Mok, Nicholas C. Wu.

**Investigation:** Weiwen Liang, Timothy J. C. Tan, Yiquan Wang, Huibin Lv, Yuanxin Sun.

**Methodology:** Weiwen Liang.

**Project administration:** Nicholas C. Wu.

**Supervision:** Roberto Bruzzone, Chris K. P. Mok, Nicholas C. Wu.

**Visualization:** Timothy J. C. Tan, Yiquan Wang.

**Writing – original draft:** Weiwen Liang.

**Writing – review & editing:** Roberto Bruzzone, Chris K. P. Mok, Nicholas C. Wu.

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
