## [Decision Letter · Decision Letter 0]

9 Jun 2022

Dear Dr Wu,

Thank you very much for submitting your manuscript "Egg-adaptation pathway of human influenza H3N2 virus is contingent on natural evolution" for consideration at PLOS Pathogens. As with all papers reviewed by the journal, your manuscript was reviewed by members of the editorial board and by several independent reviewers. In light of the reviews (below this email), we would like to invite the resubmission of a significantly-revised version that takes into account the reviewers' comments.

We cannot make any decision about publication until we have seen the revised manuscript and your response to the reviewers' comments. Your revised manuscript is also likely to be sent to reviewers for further evaluation.

Sincerely,

Florian Krammer, PhD

Associate Editor

PLOS Pathogens

Mark Heise

Section Editor

PLOS Pathogens

Kasturi Haldar

Editor-in-Chief

PLOS Pathogens

orcid.org/0000-0001-5065-158X

Michael Malim

Editor-in-Chief

PLOS Pathogens

orcid.org/0000-0002-7699-2064

Reviewer's Responses to Questions

**Part I - Summary**

Reviewer #1: Liang et al. complete a nice set of experiments analyzing the compatibility of different egg-adaptative HA substitutions in different H3N2 backgrounds. Wu and colleagues recently demonstrated that G186V and L194P substitutions in HA are incompatible, and the current study expands this to examine additional substitutions in other HA backgrounds. They first identify different egg-adaptive substitutions in different egg based H3N2 vaccine strains. They then study replicative fitness (or ability to rescue) viruses with different combinations of HA mutations. They ultimately identify epistatic interactions, including those involving key egg/antigenic substitutions such as T160K, that are in these vaccine strains, and conclude that different egg-adaptations are favored in different genetic backgrounds.

Reviewer #2: In this manuscript by Liang et al., the authors sought out to understand how egg adaptations of H3N2 differ across distinct H3N2 viruses. The authors nicely show epistasis of key residues associated with egg mutations and how this relates to distinct H3N2 clades. The manuscript is very well written and the topic is important. The manuscript can be hard to follow at times, largely due to the massive number of mutations being studied. Simplification of the key residues and relationship between these as a panel somewhere in the paper would greatly benefit the presentation of this manuscript. Otherwise, all my comments are quite minor and easily addressable.

Reviewer #3: The authors assess how natural mutations in seasonal H3N2 virus impact on the adaptation of these same viruses when cultured in eggs. H3N2 viruses are frequently associated to a higher mismatch of current seasonal vaccines in comparison to group 1 and influenza B viruses. One important aspect to consider here is that most currently licensed influenza vaccines are manufactured in eggs. Therefore, it is of interest to understand if changes in antigenicity of H3N2 viruses when passaged in eggs are related to specific DNA/protein sequences of circulating H3N2 influenza viruses. The manuscript is well-written and conveys a clear message to the audience. There are some minor points that should be addressed before acceptance.

Reviewer #4: In this manuscript Liang et al, perform a series of experiments to elucidate the trajectory and limitations of the mutation (amino acid variations) that arise in the hemagglutinin (HA) protein of H3N2 human seasonal viruses during egg-adaptation. In previous studies they identified the amino acid change L194P as an important egg-adaptation that often occurs during passaging of human H3N2 in eggs. Herein they identify specific amino acid changes that appear to be compensatory mutations that are generated in the presence of amino acid P194 in in HA. The authors also identify two naturally occurring amino acids (T160 and N190) that are appear to be incompatible with mutation P194, suggesting a potential structural restrictions on accommodating some of these residue variations. The results provide relevant information on egg-adaptations that impact vaccine development. Nonetheless, there are a few issues that need to be addressed to improve the clarity of the manuscript, in particular on how the authors are assessing and interpreting fitness.

**Part II – Major Issues: Key Experiments Required for Acceptance**

Reviewer #1: 1. Overall, the study is very nice and the experiments are well control and the data are solid. There are some parts of the manuscript where I felt ‘a bit in the weeds’ trying to follow along with different combinations of substitutions, but the overall experimental question is important and the data support the idea that different egg adaptations are favored in different genetic backgrounds.

2. The study could be strengthened if there was a better structural/mechanistic explanation for some of the findings. For example, Wu and colleagues showed earlier that RBS height explained the incompatibility of L194P and G186V. The authors of the current study show that is not the case in panel Fig 3F for incompatibility of N190 and P194, but additional structural experiments could address the mechanism further about this incompatibility.

3. The authors conclude that the specific types of egg-adaptations that arise is complicated and dependent on the evolving H3N2 strains. It would be useful to be able to predict specific egg-adaptations based on sequence—is this impossible at this point or could preliminary predictive models be made?

Reviewer #2: N/A

Reviewer #3: (No Response)

Reviewer #4: - It is unclear how the authors assessed replication fitness of the rescued viruses. In the methods they mention that after transfection the viruses were harvested and inoculated into hMDCKs, and then the supernatant were harvested once CPE was observed. These supernatants (containing the viruses) were then titrated by a TCID50 assay. Hence, it seems that the authors are determining fitness as a measure ability to rescue viruses and the generation of progeny virus (titer) after CPE was observed.

Was there any standardization to control the time that each of the viral infections took to generate CPE?; e.g. Do you collect the supernatant at the first sign of CPE? After a few hours of CPE? Until all the cells show CPE?

- To determine differences in replication fitness, one would have to perform a side-by-side growth curve for 48 hrs and plot them as such to determine and compare the kinetics of replication. As this is a major methodology of the paper (shown in Figs 1B, 2B and E, and 3B-D), these experiments need to be explained in more detail to understand how the authors are controlling for time differences that the cultures take to determine CPE, and the time when the sups were collected. It will also be valuable to actually show growth curves in cases where the growth kinetics are in fact slower (or faster).

- Can you comment on the number of rescue attempts that were done for each of the mutant viruses? After how many attempts did the authors determined that a virus could not be rescued? How were the rescue attempts controlled? Please also clarify if all viral stocks were sequenced after rescue and prior to passage in eggs, to confirm the sequence integrity of each rescue. Include that information in detail in the methods section.

- For the figure legends 1B, 2B and E, and 3B-D, provide more information on the experiment? e.g. since the results of the rescue attempts in the 293T/hMDCK… For expalme in Line 356, “…in a virus rescue experiment in mammalian cells (293T/hMDCK) after XX attempts”. Indicate what the error bars are (e.g. Line 359: SD of three experiments represented by the error bars).

- The use of “egg-adaptation pathway” is a bit confusing, as this would imply a unique way in which the mutations might generate. The data shown (e.g. Figs 2, 3 and 4) indicate that different mutations (amino acid variations) in deed occur depending on the genotype of the initial human H3N2 strain. Hence, it appears that alternative amino acids (e.g. more then one residue) can emerge at different positions as compensatory mutations. Therefore, I suggest revising the use of “egg-adaptation pathway” throughout the manuscript, including the tittle, to better reflect the conclusions of the study. For example, a tittle a long these lines: “Egg-adaptation compensatory mutations of human influenza H3N2 virus are dependent on natural evolution”, might be more aligned with the results shown.

**Part III – Minor Issues: Editorial and Data Presentation Modifications**

Reviewer #1: It might be nice to add details about the new 2022-23 Northern Hemisphere H3N2 vaccine strain (and current SH vaccine) which doesn’t have the N158 glycosylation site.

Reviewer #2: 1. At the end of figure 1 please include a phylogenetic tree of the different viruses being tested and the mutations associated with which. This will help clarify how the egg-adaptation and H3N2 strains relate. Alternatively, a structure of HA and which residues are associated with each strain would be useful.

2. Line 150, please clarify this is circulating virus and not egg-grown virus.

3. Line 175, can the authors expand on why RBS height may matter? I’d assume this obstructs receptor binding, but it’s not clear.

4. Line 215 – should read “which all belong to clade…”

5. The authors do a lot of virus rescuing but it’s not explicitly clear how they did so. They seem to use reverse genetics to generate their viruses and to rescue them in MDCK cells and then use eggs to show stability of mutations. Clarity on methods are appreciated.

Reviewer #3: Line 95-97) What do authors mean by background mutations? Are these egg mutations having a deleterious impact on the antigenicity of the virus?

Line 150-151) What do authors mean by “T160, which introduced an N-glycosylation site at N158”? Does this N-glycosylation occur due to a higher accessibility to this site of glycosyltransferases after the T160 mutation takes place?

Line 226-227) Do authors mean that if for instance there is a prevalent circulating H3N2 strain but it is likely that when growing it in eggs there appears remarkable antigenic changes, it is better to select another circulating strain, though less prevalent in humans, but less prone to antigenic changes in eggs? Of course we are not considering here the use of mammalian cell lines to grow these viruses, we assume only that the egg platform is the only possibility.

Discussion point) Is the rationale of this study based on the egg-adapted mutations discussed in this study since they are the most prevalent seen in H3N2 strains? Are there other relevant mutations not mentioned in this study?

Discussion point) according to the data provided by the authors, it seems the deleterious mutation L194P is less prevalent in egg-passaged H3N2 strains. Do authors believe that mismatches to current circulating H3N2 strains due to egg-passaging are less likely to occur? Discussion on this topic would be interesting for the audience.

Reviewer #4: 1. In the model on Figure 3, for orientation it would be helpful to include/highlight the locations of residues 160 and 98.

2. In the Figure 4 legend, include the method and bootstrap used to construct the phylogenetic tree and whether the tree was rooted or not.

3. Line 52; change to “which is the prototypical receptor for avian influenza viruses”

4. Line 52; Change to “Since the HA RBS partial…”

5. Line 62; should be “discovered”

6. Line 153; is this supposed to be 2020 and not 2021??

7. Line 155; it should be “titers”

8. Line 157-158; change to “…as determined in the egg-adapted viruses”

PLOS authors have the option to publish the peer review history of their article (what does this mean?). If published, this will include your full peer review and any attached files.

Reviewer #1: No

Reviewer #2: No

Reviewer #3: No

Reviewer #4: No
---

## [Decision Letter · Decision Letter 1]

12 Sep 2022

Dear Dr Wu,

We are pleased to inform you that your manuscript 'Egg-adaptive mutations of human influenza H3N2 virus are contingent on natural evolution' has been provisionally accepted for publication in PLOS Pathogens.

Best regards,

Florian Krammer, PhD

Associate Editor

PLOS Pathogens

Mark Heise

Section Editor

PLOS Pathogens

Kasturi Haldar

Editor-in-Chief

PLOS Pathogens

orcid.org/0000-0001-5065-158X

Michael Malim

Editor-in-Chief

PLOS Pathogens

orcid.org/0000-0002-7699-2064

Reviewer Comments (if any, and for reference):

Reviewer's Responses to Questions

**Part I - Summary**

Reviewer #1: (No Response)

Reviewer #2: The authors have addressed my concerns.

Reviewer #3: (No Response)

**Part II – Major Issues: Key Experiments Required for Acceptance**

Reviewer #1: (No Response)

Reviewer #2: N/A

Reviewer #3: (No Response)

**Part III – Minor Issues: Editorial and Data Presentation Modifications**

Reviewer #1: (No Response)

Reviewer #2: N/A

Reviewer #3: (No Response)

PLOS authors have the option to publish the peer review history of their article (what does this mean?). If published, this will include your full peer review and any attached files.

Reviewer #1: No

Reviewer #2: No

Reviewer #3: No

---

## [Editor Report · Acceptance letter]

20 Sep 2022

Dear Dr Wu,

We are delighted to inform you that your manuscript, "Egg-adaptive mutations of human influenza H3N2 virus are contingent on natural evolution," has been formally accepted for publication in PLOS Pathogens.

Best regards,

Kasturi Haldar

Editor-in-Chief

PLOS Pathogens

orcid.org/0000-0001-5065-158X

Michael Malim

Editor-in-Chief

PLOS Pathogens

orcid.org/0000-0002-7699-2064